# Reservoir Operation Policy based on Joint Hedging Rules

**Baohui Men \*, Zhijian Wu, Yangsong Li and Huanlong Liu**

Beijing Key Laboratory of Energy Safety and Clean Utilization, Renewable Energy Institute, North China Electric Power University, Beijing 102206, China; 1172211088@ncepu.edu.cn (Z.W.); lys18811551843@163.com (Y.L.); liuhuanlongHD@163.com (H.L.)
**\*** Correspondence: menbh@ncepu.edu.cn; Tel.: +86-010-6177-2451

**Abstract:** When the water supply capacity of the reservoir is small, hedge rule (HR) can be applied to reduce the risk of unacceptably large damage from water shortage during drought. Moreover, in water-receiving areas of water diversion project, it is important to reduce transfer based on HR when the water-receiving area is in a wet period so as to reduce the water transfer cost. This paper improved the traditional HR and proposed a new kind of hedging rule named joint hedging rule (JHR). JHR was applied to Yuqiao Reservoir of Tianjin in China and was compared with HR and standard operation policy (SOP) as two control groups. The result indicates that JHR performs better than HR and SOP, which cannot only mitigate the risk of unacceptably large damage from water shortage by one hedge process but also reduce the transferred water by another hedge process. In addition, the number of days of different water shortage, the storage ratio at the end of the year, and transferred water result indicates that JHR is of high reliability and practicability.

**Keywords:** reservoir operation policy; joint hedging rule; water shortage; transferred water

## 1. Introduction

Over the past decades, solving reservoir operation problems has been a challenge for water resource planners and managers. For optimum operations, several rules, as well as many simulations and optimization models, have been presented, such as real option model, linear decision rule, pack rule, space rule, standard operation policy (SOP), hedge rule (HR), and so on. The SOP [1], which is the simplest and most widely used one, requests the release in each period of the demand, if possible. However, because of the whole available water supplied in the current period, if the inflow in the following periods is small, large damage from water shortage will take place. This means the SOP frequently induces a high-percentage single-period deficit [2]. Thus, a considerable number of studies have proposed various hedging mechanisms modifying the SOP [3–6]. The hedging theory of economics is applied to the reservoir operation policy during drought to mitigate a high-percentage single-period deficit.

During periods of drought or when anticipating a drought, it may not be possible to reach the storage target because of an inflow shortage. To minimize the impact of drought and the consequent shortages in current and future water supply, the hedging rules are used to balance the shortage in water supply with the storage target [7,8]. As the relationship between the damage and the shortage is nonlinear, by reducing the supply of reservoir when supply capacity is small [9], HR can store some water in the reservoir for future usage and reduce the possibility of large shortage damage from water shortage in future [10–12]. HR and SOP are shown in Figure 1.

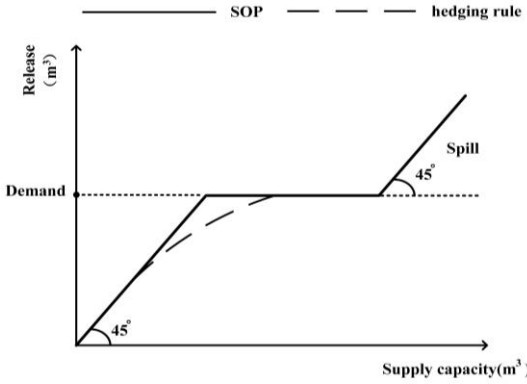

**Figure 1.** Curves of HR and SOP.

According to Figure 1, during normal periods of operation, all planned demands are met at the 100% level, and the reservoir storage is kept at or above the established target storage level. However, during periods of drought, when the reservoir storage falls, the reservoir release is reduced to ensure that a sufficient amount of water remains in storage for future water supply. This situation creates a trade-off between meeting ongoing demands and maintaining adequate reservoir storage when inflow is insufficient [13]. Using the hedging rules along with the rule curves, a balance can be achieved between the water supply shortage and the storage objectives.

Several studies on the hedging rules have been reported in the literature. Tscheikner developed a continuous linear hedging rule and converted the continuous hedging rule into multiple discrete hedging rules. Besides optimizing the parameters of a curve with a shape defined a priori, Sangiorgio adopted an artificial neural network and searched for its best parametrization [14]. Then, a multiple hedging model was developed by using mixed integer programming [15]. Different hedging rules result in different water-shortage characteristics, which thus can indicate the effects of hedging and are used to derive the optimal hedging rules. Shortage indicators include minimization of the sum of squared deficits, minimization of maximum single-period [13], minimization of drought-duration-weighted shortage [12], and minimization of the weighted average of two shortage indicators [16]. Shortage indicators generally are inconsistent and may be conflicting. That is, a hedging rule may improve one shortage indicator but at the cost of deteriorating another shortage indicator. In addition, hedging parameters, which include onset, termination of hedging, and the percentage of water rationing that varies with time, are difficult to be determined.

Therefore, many types of research have been done to get the optimal hedging rule. Some researchers proposed a new objective function or new index to get the optimal hedging rule. Tatano [17] proposed a new index which is closely related to both the length of shortage period and the shortage amount. Neelakantan and Pundarikanthan [18] took the minimum sum of squares of shortage ratio in each period as the objective function. Shiau and Lee [19] proposed that HR has the risk of increasing the whole water shortage and proposed objective function which can reduce the whole shortage and the maximum shortage in each period. Ji et al. [12] defined shortage damage depth, used HR to reduce the damage depth and researched how to get optimal parameters of HR. Some researched the ways many factors would influence the performance of HR or in what form HR would have the best performance. You and Cai [20] researched how the factors, such as the runoff uncertainty and the reservoir's storage capacity, would influence the performance of HR. Srinivasan and Philipose [21] proposed the value function of supplying and storing. Based on the optimal hedging section theory (the optimal point to start or end hedging is when the marginal value of storing is equal to that of supplying), the optimal hedging section in each period could be found. Draper and Lund [22] researched how the function form of storing or supplying water would affect the performance of HR and proposed that probably the quadratic function is the best function form. They also researched how hedge form (such as a single-point hedge, two-point hedge, linear hedge, and non-linear hedge) would affect hedge

performance and put forward that a linear two-point hedge may be the optimal hedge form. Some researchers combine HR with a new theory. Xu et al. [9] used a risk-averse criterion to rationalize water supply to overcome the shortcomings of risk-neutral hedging rule in minimizing water shortage impacts in unfavorable realizations, in which actual inflow is less than anticipated. Hu et al. [23] proposed a novel hedging rule to improve the water supply efficiency of a reservoir, in which, hedging process contains three different hedging sub-rules through a two-step approach. Others enlarged the application scope of the hedging theory or improved the hedging process, including limiting the supply of different requirement in turn according to the varied importance of agricultural, industrial, and domestic purpose with the decreasing supply capacity [24]. Releasing before a flood can increase reservoir storage capacity available to capture more severe flood flows and developed an optimization model for pre-storm flood hedging releases [25].

In the previous study of hedging rules in water-receiving areas, usually the transferred water flows into large local reservoirs, and by limiting the water supply of reservoir, the hedging rule can reduce the risk of large shortage in future (only water shortage caused by insufficient water supply was considered). In fact, the transferred water does not flow into large local reservoirs. In this case, if the transferred water or the local reservoir supply is too small, there will be a water shortage caused by small water distribution network capacity, water purification capacity. At present, there is no research to hedging rules in the situation when the transferred water does not flow into large local reservoirs. In response to this situation, this paper improved HR and proposed a new kind of hedging rule named joint hedging rule (JHR) and researched the lower boundary of JHR's hedge rate which would cause a water shortage. A bi-level optimization model was proposed, and an improved particle swarm optimization algorithm was applied to find the optimal hedge process of JHR. The traditional hedging rule is to limit the water supply when the water supply is small, so as to save the water for future use and reduce the risk of serious water shortage events. For the external water transfer area, the cost of water diversion is high due to external water transfer. In order to reduce the cost of water transfer, the novelty of JHR proposed in this paper is that compared with the traditional hedging rule, when the local reservoir has a large amount of water available, JHR changes the water supply of external water and local water. The composition of the local reservoir water supply can maintain the local reservoir water storage in a better interval, reduce the water abandonment caused by excessive water storage, and reduce the water transfer cost. JHR has been applied to representative Tianjin case, and HR and SOP are used as a baseline to evaluate the JHR performances. The performance indicates that JHR can reduce the damage from shortage and increase the reservoir's storage at the end of the year as well as reduce transferred water.

## 2. Materials and Methods

### 2.1. The Expression form of Hedging Rules

Hedge process can be defined by SWA (starting water availability), EWA (ending water availability), and hedge rate (limit ratio of water supply). SWA and EWA are the maximum and the minimum supply capacity (storage + inflow) of hedge process. This means that when the water storage above the dead water level of the reservoir is in SWA or EWA, the hedging rules can be effective. Hedge rate, also called limit rate, can define the limit supply process. Generally, single-point hedging rule and two-point hedging rule are the most common HR. The hedging process triggers are in the section between SWA and EWA. EWA is the coordinate origin for single-point hedging rule. However, EWA isn't the coordinate origin as for two-point hedging rule, which is a continuous hedging rule. Single-point hedging rule, two-point hedging rule, three-point hedging rule are shown in Figures 2 and 3.

Compared with HR, JHR has one more hedge process. $SWA_1$, $EWA_1$ and $SWA_2$, $EWA_2$ are, respectively, SWA, EWA of JHR's two hedge process, and the reservoir storage can be divided into five sections by them. The five sections are shown in Figure 4, and the process of JHR is shown in

Table 1. Section 5 is an abandoned section, in which the storage is larger than normal high-water level or flood limit level. When the supply capacity of the reservoir is in Section 4, the supply capacity of the reservoir is sufficient, and hedge rule can be applied to increase reservoir's supply and reduce transfer, and in this section, the single-point hedging rule is chosen as the basic form. When the reservoir's supply capacity is in Section 2, in order to reduce the possibly unacceptably large damage from shortage, another hedge process would reduce the supply of local reservoir and increase transfer. But the supply of local reservoir shouldn't be too small so as not to cause water shortage due to small water distribution network capacity, water purification capacity, and so on. Similarly, the transferred water shouldn't be too small too. Thus, compared with traditional HR, JHR's two hedge rate shouldn't be too small, so as not to cause water shortage, we made the two smallest hedge rate $p_e$ and $q_s$.

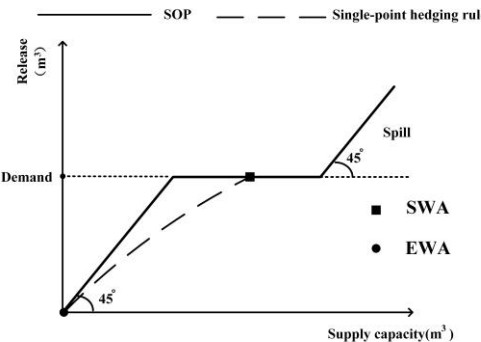

**Figure 2.** Scheme of one-point hedging rule.

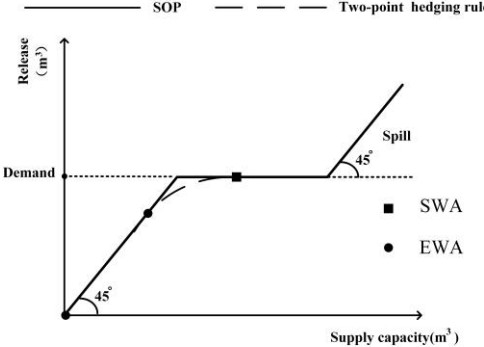

**Figure 3.** Scheme of two-point hedging rule.

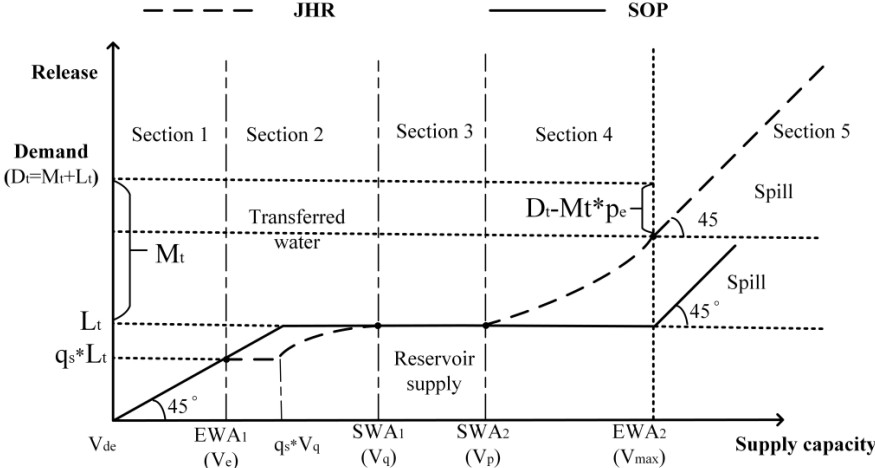

**Figure 4.** Comparison of curves of JHR and SOP.

**Table 1.** Supply process of JHR.

|  | Scope | Transfer | Reservoir | Hedge Rate |
|---|---|---|---|---|
| Section 1 | $V_{de} < V_t \leq EWA_1$ | $D_t - L_t \times q_s$ | *supply capacity* | |
| Section 2 | $EWA_1 < V_t \leq SWA_1$ | $D_t - q \times L_t$ | $q \times L_t$ | $q \in (q_s, 1)$ |
| Section 3 | $SWA_1 < V_t \leq SWA_2$ | $M_t$ | $L_t$ | |
| Section 4 | $SWA_2 < V_t \leq V_{max}$ | $M_t \times p$ | $D_t - M_t \times p$ | $p \in (p_e, 1)$ |
| Section 5 | $V_t > V_{max}$ | $M_t \times p_e$ | $D_t - M_t \times p_e$ | |

where the $D_t$ is the whole demand needed to be satisfied by the transferred water and reservoir in $t$ period; $M_t$ and $L_t$ are the part which needed to be satisfied by the reservoir and the transferred water in $t$ period ($D_t = M_t + L_t$); $p$ and $q$ are hedge rate, their values are, respectively, from 1 to $p_e$ ($p_e < 1$) and $q_s$ to 1, and $p_e$, $q_s$ are the smallest hedge rate of the two hedge process. $p$, $q$ can be acquired by Equations (1)–(3).

When the supply capacity is between the $V_{max}$ and $V_p$:

$$p = p_e + (1 - p_e)(V_{max} - V_t)/(V_{max} - V_p), \tag{1}$$

When the supply capacity is between $V_q$ and $q_s \times V_q$:

$$q = 1 - (V_q - V_t)/(V_q - V_{de}), \tag{2}$$

When the supply capacity is between $q_s \times V_q$ and $V_e$:

$$q = q_s, \tag{3}$$

where the $V_{max}$ is the storage of reservoir's flood limit level in flood season or the storage of reservoir's normal high-water level in non-flood season; $V_t$ is the supply capacity in $t$ period; $V_p$ is the storage of $SWA_2$; $V_q$ is the storage of $SWA_1$; $V_e$ is storage of $EWA_1$.

*2.2. Method Solution and Analysis*

JHR has two hedge process. After the upper hedge process is determined ($SWA_2$ and $EWA_2$), the lower hedge process would search its optimal decision ($SWA_1$ and $EWA_1$). In other words, JHR's two hedge process fit within the framework of leader-follower or Stackelberg game. A bi-level optimization model and an improved particle swarm optimization algorithm [26] can be applied to solve this kind of problem. As a shuffling process is introduced in the improved particle swarm optimization algorithm, it performs better than the particle swarm optimization algorithm in the following aspects: first, it can ensure information sharing between subgroups and the overall group; second, it has less possibility of encountering local optima; third, it can avoid precocious phenomena to a certain extent. This paper proposed a bi-level optimization model, and an improved particle swarm optimization algorithm was applied. The external loops number ($T = 100$), the loops number of the bi-level optimization model is 50. The flowchart of solving the bi-level optimization model for JHR is shown in Figure 5.

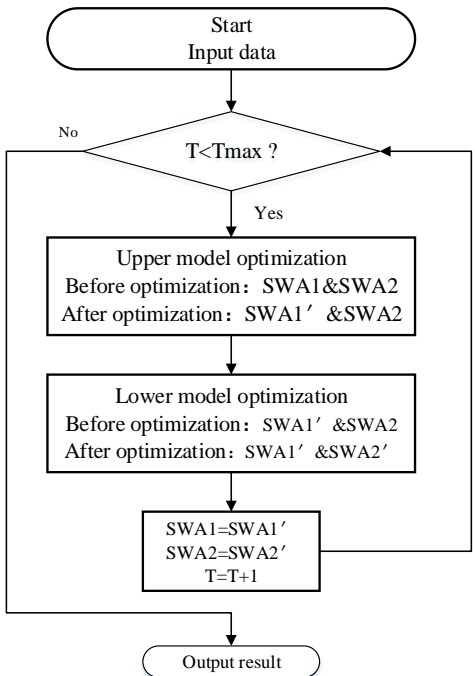

**Figure 5.** The flowchart for solving a bi-level optimization model.

### 2.3. The Objective Function and Constraints

The bi-level optimization model contains two objective functions: the upper one and the lower one. The upper objection function is to maximize the value of the reservoir's storage ratio at the end of the year minus the whole transfer in the year. The lower objective function is to minimize the sums of squares of shortage ratio in each period [24]. The upper and lower models compete with each other to find the optimal solution. The optimal solution is obtained in the upper model, which is used as the input of the lower layer model. Then, the optimal solution is substituted into the upper model in the lower model, and the cycle is calculated back and forth until the condition of the optimal solution is satisfied. The two objective functions and constraints are shown as follows:

The upper objective function:

$$Max\ K = \alpha \times m - \sum_{t=1}^{T} L_t, \tag{4}$$

The lower objective function:

$$Min\ S = \sum_{t=1}^{T} \left( \frac{D_t - M_t - L_t}{D_t} \right)^2, \tag{5}$$

The constraints are shown as follows:

1. Transfer no more than the supply capability or the transfer capacity of the water transfer project

$$L_t \leq L_{mt},\ L_t \leq L_l \tag{6}$$

2. Largest and smallest storage constraints of the reservoir

$$V_{de} \leq V_t \leq V_{max}, \tag{7}$$

3. Water balance constraint

$$V_t = V_{t-1} + x_t - M_{t-1} - Q_{It}, \tag{8}$$

4.    Non-negative constraints

$$D_t \geq 0;\ L_t \geq 0;\ x_t \geq 0;\ m \geq 0;\ \alpha \geq 0, \tag{9}$$

where the $x_t$ is the runoff of t period; $L_{mt}$ is the supply capacity of the transferred water in t period; $m$ is the reservoir's storage ratio at the end of year; $\alpha$ is a preset parameter between 0 and 1, represent the different importance of transfer and reservoir's water storage at the end of the year (in the considered case, $\alpha$ is 0.5); $L_l$ is the transfer capacity of the water diversion project.

## 2.4. Study Area

Tianjin is a city which seriously lacks water. Yuqiao Reservoir sits on the north of Tianjin. The South-to-North Water Diversion Project (SNWDP) is a large water transfer project, which can transfer the water from the Hanjiang River to Tianjin in order to alleviate the shortage of water. The trunk canal of SNWDP passes through four provinces and municipalities under the central government of Henan, Hebei, Beijing, and Tianjin. It provides water for livelihood, industrial, and agricultural production for more than a dozen big and medium-sized cities along the line. Tianjin, SNWDP, and Yuqiao Reservoir are shown in Figure 6.

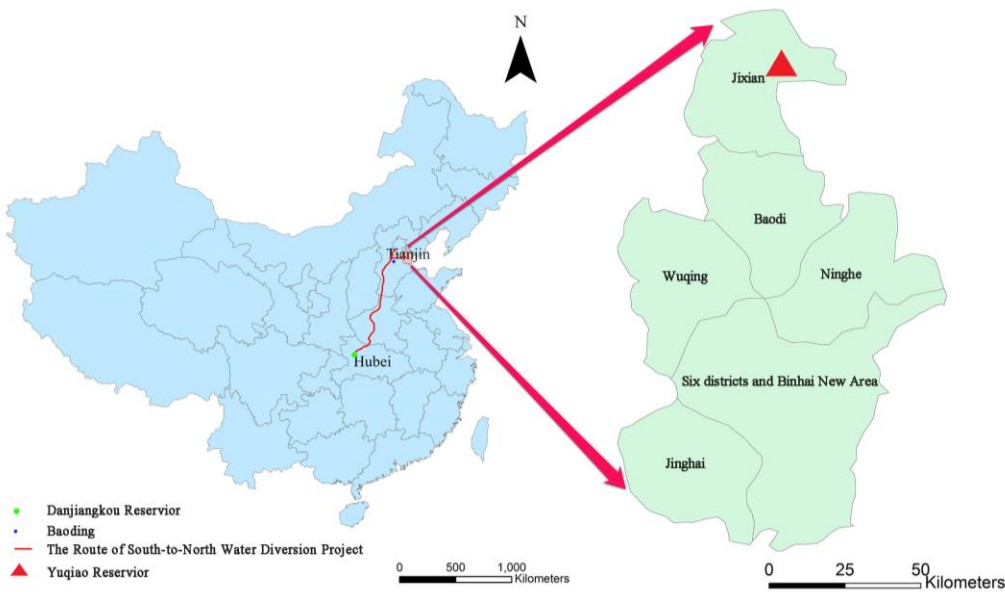

**Figure 6.** Location of SNWDP, Yuqiao Reservoir, and Tianjin in China.

Tianjin's water sources include surface water, groundwater, Luanhe-Tianjin water, diversion water from the Yellow River, water from the SNWDP, desalinated seawater, and reclaimed water. Since the Yellow River Water Diversion Project belongs to backup water sources, Luanhe-Tianjin water flows into Yuqiao Reservoir, which has a large regulation capacity. The average annual transfer of the South-to-North water is about one-third of the average annual demand of Tianjin, and thus the South-to-North water would greatly influence the satisfaction degree of Tianjin's water supply system. There is no large reservoir in Tianjin that can accept transferred water. Tianjin's urban water supply network is shown in Figure 7.

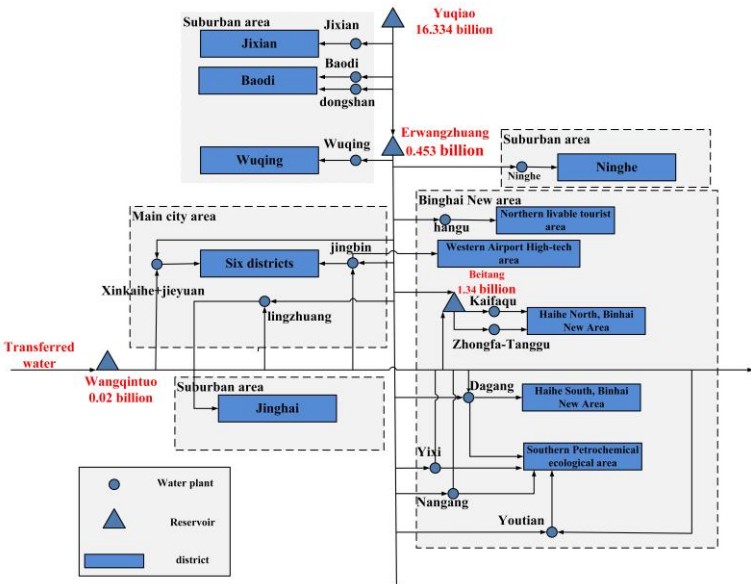

**Figure 7.** Tianjin's urban water supply network.

## 3. Result

Li [27] researched through SNWDP that how much water could be transferred from the Danjiangkou Reservoir (the source of SNWDP) from 1956 to 1997. The conclusion is that, in an extremely dry year, the amount of water which can be transferred is quite smaller than that of a dry year. However, limited by the project scale of the SNWDP, the amount of water which can be transferred is relatively close in a dry year and wetter year. The sums of squares of daily shortage ratio of Tianjin in the case of ($d$, $xd$), the storage ratio of Yuqiao Reservoir at the end of the year, and the transfer in the case of ($m$, $d$) are researched in this paper. $w$, $m$, $d$, $xd$ represent the water-receiving area or the water-transferred area in a wet year, normal year, dry year, and extremely dry year, respectively. The Pearson type III curve is used in hydrological frequency calculation. The wet year, normal year, dry year, and the extremely dry year has an insurance rate of 25%, 50%, 75%, and 95%, respectively. The ($m$, $d$), for example, represents for the water-receiving area and water-transferred area in a normal year and dry year, respectively.

This paper collected daily runoff data of Yuqiao Reservoir and the Danjiangkou Reservoir. As for SWNDP, according to Proximity Principle in runoff data, 1981 and 1987 were a dry year and extremely dry year, respectively. 1998, 1984, 1982, and 2000 were Yuqiao Reservoir's wet year, normal year, dry year, and extremely dry year, respectively. The water demand is the average of the water consumption in Tianjin in the past ten years. The flooding period of the water-receiving area has been kept in the front, JHR's rule curves are shown in Figure 8. (The rule curves reflect that in dry period of the year (from day 149 to day 344), the storage volume should be high, and in wet seasons, the storage volume of the reservoir can be lower).

Two water supply rules of the reservoir are settled: HR and SOP. As for HR, when the water supply capacity is smaller than SWA, reduced reservoir's supply and the reset demand would be satisfied by transfer. Even if the transferred water isn't sufficient for the rest demand, the reservoir wouldn't supply any more. The performance of HR is studied as the smallest limit rate changes between 0 and one. The optimal range of hedge rate is from $0.5M_t$ to $M_t$. The sum of squares of daily water shortage ratio, transfer, and the storage ratio at the end of the year of SOP, HR, and JHR in each case are shown in Tables 2–4.

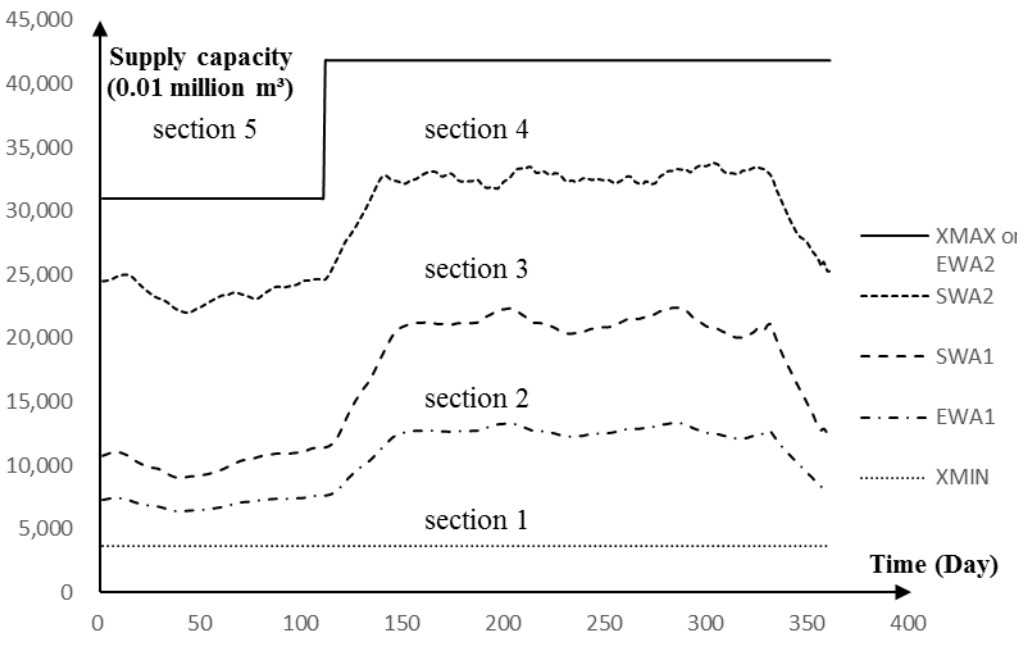

**Figure 8.** The reservoir operation policy based on JHR.

It can be seen from the Table 2 that the hedging rules can play a good role mainly when the water-receiving area is in extreme dry year. Especially under the circumstance of (*xd*, *xd*), the Yuqiao Reservoir provides more water based on SOP, but sums of squares of daily shortage ratio based on JHR and HR are 2.35 and 2.39, respectively, less than that based on SOP. This proves that the hedging rule can effectively reduce the water shortage rate per unit time when the water supply of the reservoir is small and avoid extreme water shortage. When the water-transferred area is in a dry year, there is almost no water shortage regardless of the circumstances of the water-receiving area. In the case of (*xd*, *d*), sums of squares of daily shortage ratio of HR is 0.22 while it is 0 for JHR. In addition, the storage ratio of the Yuqiao Reservoir at the end of the year is larger under the condition of JHR than HR, indicating that the Yuqiao Reservoir will have more water available in the next year. Generally speaking, the SOP rule is better than HR under non-extreme water shortage conditions. Because the water supply is enough in this case, the SOP can ensure that no water shortage occurs, and HR still restricts the water supply and causes the water shortage event. JHR proposed in this paper solve this problem well. When the water-receiving area is not deficient in water under SOP conditions, there is no shortage of water under JHR.

In order to more intuitively express the advantages of JHR relative to HR and SOP, Figure 9 is drawn on the basis of sums of squares of daily shortage ratio and the amount of transferred water. Because there is almost no water shortage when the water-transferred area is dry and the amount of transferred water is same when the water-transferred area is extremely dry, this situation is shown in the figures. As can be seen from Figure 9a, when the water-transferred area is extremely dry and water-received area is dry or extremely dry, JHR and HR can reduce the sums of squares of daily shortage ratio. That is to say, JHR and HR are effective and can avoid extreme water shortage under the condition of (*xd*, *xd*) and (*d*, *xd*). This also proves that the hedging rules are only applicable in the dry season. As for the difference of JHR and HR, Figure 9b shows that JHR rules can use less transferred water to reduce water transfer costs under the same conditions of sums of squares of daily shortage ratio.

**Table 2.** Sums of squares of daily shortage ratio, transfer, and the storage ratio at the end of the year of JHR.

| Cases | *xd, xd* | *d, xd* | *m, xd* | *w, xd* | *xd, d* | *d, d* | *m, d* | *w, d* |
|---|---|---|---|---|---|---|---|---|
| Sums of squares of daily shortage ratio | 49.07 | 30.14 | 12.36 | 21.38 | 0.00 | 0.00 | 0.00 | 0.00 |
| Transfer (0.1 billion) | 5.86 | 5.86 | 5.86 | 5.86 | 10.42 | 9.17 | 8.87 | 8.95 |
| Sum of Yuqiao supply (0.1 billion) | 3.9 | 5.11 | 6.75 | 5.86 | 3.54 | 4.8 | 5.09 | 5.02 |
| Storage ratio | 0.22 | 0.30 | 0.32 | 0.32 | 0.33 | 0.38 | 0.85 | 0.61 |

**Table 3.** Sums of squares of daily shortage ratio, transfer, and the storage ratio at the end of the year of HR.

| Cases | *xd, xd* | *d, xd* | *m, xd* | *w, xd* | *xd, d* | *d, d* | *m, d* | *w, d* |
|---|---|---|---|---|---|---|---|---|
| Sums of squares of daily shortage ratio | 49.11 | 30.15 | 11.57 | 20.82 | 0.22 | 0.00 | 0.00 | 0.00 |
| Transfer (0.1 billion) | 5.86 | 5.86 | 5.86 | 5.86 | 10.38 | 10.11 | 10.04 | 10.04 |
| Sum of Yuqiao supply (0.1 billion) | 3.88 | 5.13 | 6.81 | 5.96 | 3.52 | 3.86 | 3.92 | 3.92 |
| Storage ratio | 0.22 | 0.30 | 0.30 | 0.32 | 0.34 | 0.42 | 0.90 | 0.63 |

**Table 4.** Sums of squares of daily shortage ratio, transfer, and the storage ratio at the end of the year of SOP.

| Cases | *xd, xd* | *d, xd* | *m, xd* | *w, xd* | *xd, d* | *d, d* | *m, d* | *w, d* |
|---|---|---|---|---|---|---|---|---|
| Sums of squares of daily shortage ratio | 51.46 | 31.19 | 9.12 | 19.12 | 0.00 | 0.00 | 0.00 | 0.00 |
| Transfer (0.1 billion) | 5.86 | 5.86 | 5.86 | 5.86 | 10.04 | 10.04 | 10.04 | 10.04 |
| Sum of Yuqiao supply (0.1 billion) | 4.19 | 5.65 | 7.34 | 6.55 | 3.92 | 3.92 | 3.92 | 3.92 |
| Storage ratio | 0.12 | 0.12 | 0.12 | 0.12 | 0.21 | 0.40 | 0.90 | 0.63 |

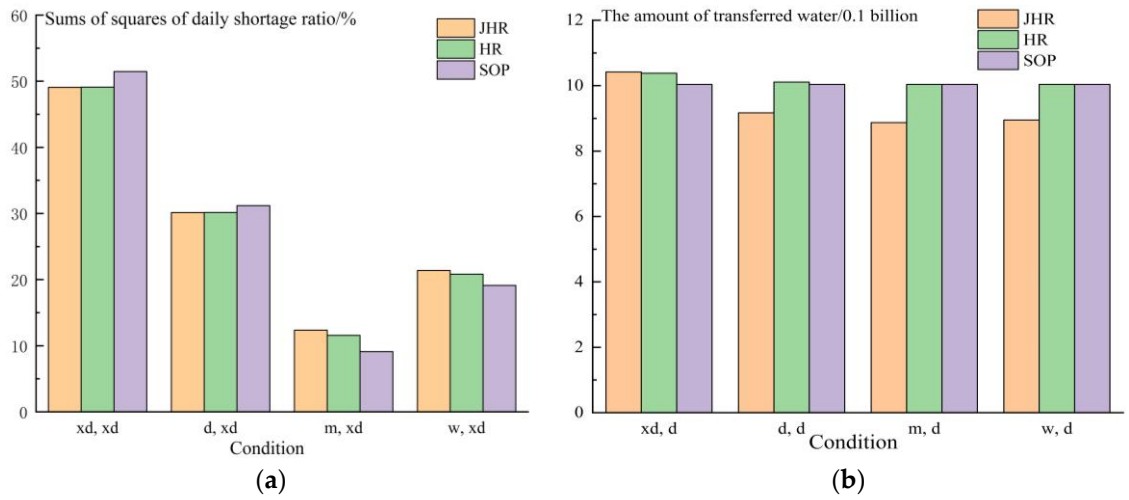

**Figure 9.** The comparative results based on JHR, HR, and SOP (**a**, **b** show the sums of the squares of daily shortage ratio and the amount of transferred water, respectively).

In order to further study the water supply situation of the external water transfer and Yuqiao Reservoir under different cases, the water supply process of Yuqiao Reservoir and transfer process in each case are shown in Figures 10–13 (Day 1 represents the beginning of the flood season, June 1st). Because when the water source area is in extreme water shortage conditions, the JHR can show better superiority than the SOP rules and HR. Therefore, this paper describes the situation when the water-transferred area is extremely dry. In the figures, zoneI represents water shortage, zoneII represents water supply by SNWDP, and zoneIII represents water supply in Yuqiao Reservoir. It can be seen from the figures that the area of zoneI decreases as the amount of water in the water-receiving area increases. The large scale of water shortage occurs in the 250th to 300th days of the year, that is, from January to March each year. The turning point of ZoneII is the beginning of limiting water

supply of Yuqiao Reservoir. When the water-receiving area is in a state of water shortage and extreme water shortage, the time limit for water supply appears earlier. Because the reservoir inflow is small in the future, in order to meet future water demand and prevent the occurrence of extreme water shortages, the water supply should be restricted earlier, so that the reservoir has relatively more water storage. Moreover, in the considered wet year of the water-receiving area, the runoff in the year is not symmetrical, about 50% of runoff comes in July, while most runoff in the flooding period turning into reservoir's surplus. Thus, the sums of squares of shortage ratio in (*w*, *xd*) case are larger than that of (*m*, *xd*).

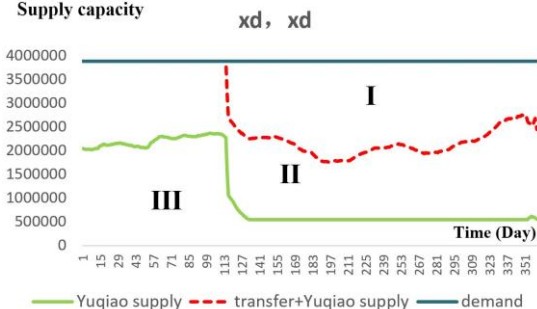

**Figure 10.** Case of (xd, xd).

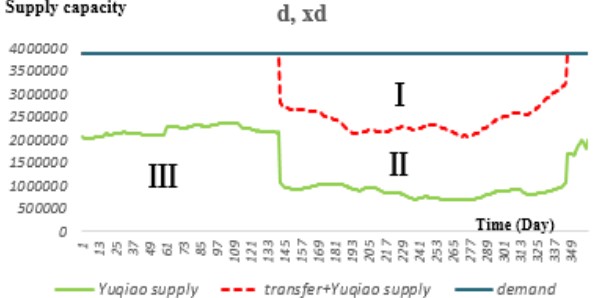

**Figure 11.** Case of (d, xd).

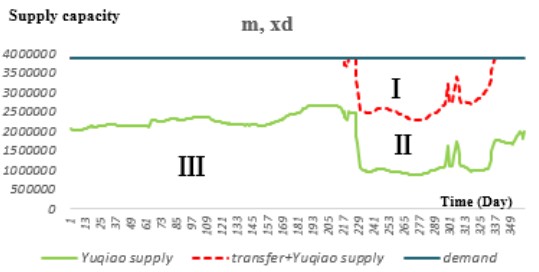

**Figure 12.** Case of (*m*, *xd*).

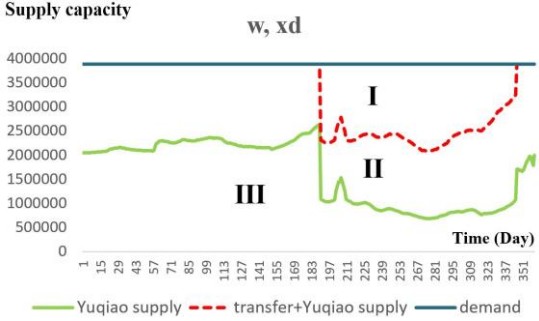

**Figure 13.** Case of (*w*, *xd*).

## 4. Discussion

As the percentage of transferred water in Tianjin's water resource is high, when the water-transferred area is in a dry year, Tianjin's water resource is insufficient for the demand and shortage occurs. Thus, a shortage occurs in the cases of (xd, xd), (d, xd), (m, xd), and (w, xd). According to HR and JHR, the reservoir's supply would be limited when the supply capacity is small, and the rest needs to be satisfied by the transferred water. Even when the transferred water isn't sufficient for the rest of the demand, the reservoir wouldn't supply anymore, thus sometimes this kind of operation policy may increase the whole shortage. For this reason, the sums of squares of daily shortage ratio and the storage ratio at the end of the year of HR and JHR are a little larger than that of SOP in cases of (m, xd), (w, xd). Though the sums of squares of daily shortage ratio of JHR and HR is a little larger than that of SOP, the number of serious shortage events (40% shortage and higher) could be smaller. The damage from a shortage can be shown by a number of days of different water shortage. The larger the number of serious shortage events is, the larger the damage is. The number of days of different water shortage in cases of (m, xd), (w, xd) are shown in Tables 5 and 6.

**Table 5.** Number of different water shortage in (w, xd) case.

| Shortage Ratio | 0%–20% | 20%–40% | 40%–60% | 60%–80% | >80% |
| --- | --- | --- | --- | --- | --- |
| SOP | 1 | 21 | 64 | 0 | 0 |
| HR | 6 | 88 | 58 | 0 | 0 |
| JHR | 5 | 88 | 56 | 0 | 0 |

**Table 6.** Number of different water shortage in (m, xd) case.

| Shortage Ratio | 0%–20% | 20%–40% | 40%–60% | 60%–80% | >80% |
| --- | --- | --- | --- | --- | --- |
| SOP | 1 | 0 | 42 | 0 | 0 |
| HR | 9 | 88 | 8 | 0 | 0 |
| JHR | 11 | 88 | 15 | 0 | 0 |

When Tianjin suffers drier cases, JHR and HR would perform better. Compared with SOP, JHR and HR have smaller sums of squares of water shortage ratio (shown in Tables 2–4). The number of days of different shortage in cases of (d, xd), (xd, xd) are shown in Tables 7 and 8.

**Table 7.** Number of different water shortage in (d, xd) case.

| Shortage Ratio | 0%–20% | 20%–40% | 40%–60% | 60%–80% | >80% |
| --- | --- | --- | --- | --- | --- |
| SOP | 1 | 28 | 110 | 17 | 0 |
| HR | 10 | 88 | 107 | 0 | 0 |
| JHR | 5 | 95 | 106 | 0 | 0 |

**Table 8.** Number of different water shortage in (xd, xd) case.

| Shortage Ratio | 0%–20% | 20%–40% | 40%–60% | 60%–80% | >80% |
| --- | --- | --- | --- | --- | --- |
| SOP | 0 | 28 | 163 | 21 | 0 |
| HR | 1 | 57 | 190 | 0 | 0 |
| JHR | 1 | 53 | 192 | 0 | 0 |

Table 7 indicates that JHR and HR can reduce the number of serious shortage events in the cases of (d, xd), and the Table 8 shows that though JHR and HR have a little larger number of serious shortage events, the damage of SOP is larger. In case of (xd, xd) and (d, xd), the number of days of large-scale water shortages (shortage ratio between 60% and 80%) on SOP reached 21 days and 17 days, respectively, while HR and JHR avoid the occurrence of such events. This reflects the

essential characteristics of the hedging rules that they are designed for conserving more water for future use during droughts by curtailing delivery even when there is enough water to meet current demand [28]. A good hedging rule can effectively reduce a very high-percentage single period shortage [29]. However, unnecessary hedging increases more frequent small shortages, and thus decreases the reliability of water supply [30]. Just as in (*d*, *xd*) case, the number of days on SOP whose shortage ratio is smaller than 40% is much smaller than that on HR and JHR. Therefore, a lot of effort has been expended in investigating the effects of hedging on water supply performance and suggesting optimal hedging rule. As for JHR, compared with HR, by increasing the supply of reservoir and reducing transfer, more local water would be supplied, reservoirs would be less likely to release surplus water, and transfer would be decreased (JHR transfer about 0.1 billion cubic meters less in the cases of (*d*, *d*), (*m*, *d*), and (*w*, *d*)). Meanwhile, the more the water is stored in the reservoir at the end of the year, the reservoir's supply capacity in the following year would be larger.

As discussed in the methodology section, water supply performance is closely affected by the storage targets. The oversized degree and range of hedging may lead to water shortage by unnecessary hedging when the value of storage target is too high; otherwise, reservoir operation guided by the hedging rule fails to store enough water to mitigate future deficits. As shown in Figures 10–13, predicted future reservoir inflows would also have a significant impact on water supply constraints. In the wet years, the inflow is large, and the reservoir does not have to increase the water storage at the expense of limiting water supply in the early stage for the future water demand. However, in extremely dry years, it is necessary to limit the water supply as early as possible to avoid catastrophic water shortages in the future. So, storage targets and inflow of the reservoir are the two main factors affecting water supply performance on JHR.

Compared with traditional HR, JHR proposed in this paper has one more hedge process and the reservoir storage divided into five sections. More critically, JHR is more meaningful for reservoirs in external water-receiving areas. As shown in Figure 4, when the supply capacity of the reservoir is in Section 4, the supply capacity of the reservoir is sufficient, and hedge rule can be applied to increase the reservoir's supply and reduce transfer, and in this section, the single-point hedging rule is chosen as the basic form. When the reservoir's water storage capacity is gradually reduced, in order to reduce the possibly unacceptably large damage from shortage, another hedge process would reduce the supply of local reservoir and increase transfer. Thus, JHR's two hedge rate shouldn't be too small so as not to cause a water shortage. JHR can make better use of external water transfer than HR, especially when there is extremely dry in the external water-transferred area and the water-receiving area. The reservoir on JHR can maintain a higher water storage ratio at the end of the dispatch period in order to increase the water supply guarantee rate for continuous dry years.

## 5. Conclusions

(1) This paper focuses on the operation policy of reservoirs in water-receiving areas. Due to the large investment and operation cost, the transferred water is much more expensive than the local water. The decision-making of rational transfer and the supply of reservoir in the water-receiving area in each period is an important problem. If there is too much transfer, transfer cost would increase, on the other hand, if there is too little transfer, the risk of large water shortage in the future would increase. This paper proposes JHR, a bi-level optimization model which increase the reservoir's storage ratio at the end of the year and reduce transfer. This paper counted up the number of days of different water shortage, which can directly compare the damage from water shortage upon the application of JHR, HR, and SOP. The number of days of different water shortage, the storage ratio at the end of the year, and the transferred water result indicate that JHR is of high reliability and practicability.

(2) The proposed JHR was applied to Yuqiao Reservoir of Tianjin in China. The considered case study of JHR, HR, and SOP in various cases indicates that JHR owns higher applicability and reliability. It is meaningful to some extent to use the sums of squares of shortage ratio to represent the

damage from water shortage, and the ongoing study will be meaningful to research the damage and the shortage precisely.

(3) To apply JHR in daily operation, the demand process, the average transferred water in each period, the reservoir characteristics, the discharge capacity of the water transfer project, runoff data of the reservoir of the water diversion project, and runoff data of the reservoir in the water-receiving area of the coming period should be given. Whether the hedging process or any other process should be applied is determined according to the section of the current water supply capacity of the reservoir.

**Author Contributions:** B.M. conceived the research theme; Z.W. provided data and designed the proposed analytical approach; H.L. performed analysis; Y.L. and Z.W. wrote the paper.

**Funding:** This work was supported by the National Key R& D Program of China (Grant No. 2016YFC0401406) and the Famous Teachers Cultivation planning for Teaching of North China Electric Power University (the Fourth Period).

**Conflicts of Interest:** The authors declare no conflict of interest.

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
