# Peer review of "Reservoir Operation Policy based on Joint Hedging Rules"

_water, doi:10.3390/w11030419_

Reviewer 1 Report

Authors should present the version before resubmission.

Some of the figures are totally illegible, it should be improved.

Manuscript is preapared in really poor way.

It should be clearly indicate, where is the novelty of the presented method.

There are a lot of publication concerning this issue.

Author Response

Dear Review:

We appreciated very much to Reviewers for your positive comments, the reviewers’ comments and suggestions are very important to improve the manuscript, and the authors thank the reviewers a lot. The relative corrections were listed below.

1. Authors should present the version before resubmission.

 Thank you for your suggestion. I have presented the version before resubmission and corrected the erroneous errors in the article.

2. Some of the figures are totally illegible, it should be improved.

 Some illegible figures have been improved in the paper (eg. Figure 4,5,10) and Figure 9 has been added.

3. Manuscript is prepared in really poor way.

 The article was read again and corrected for unreasonable expressions to make it easier for readers to understand

4. It should be clearly indicated, where is the novelty of the presented method.

 The traditional hedging rule is to limit the water supply when the water supply is small, so as to save the water for future use and reduce the risk of serious water shortage events. For the external water transfer area, the cost of water diversion is high due to external water transfer. In order to reduce the cost of water transfer, the novelty of JHR proposed in this paper is that compared with the traditional hedging rule, when the local reservoir has a large amount of water available, JHR changes the water supply of external water and local water. The composition of the local reservoir water supply can maintain the local reservoir water storage in a better interval, reduce the water abandonment caused by excessive water storage, and reduce the water transfer cost.

5. There are a lot of publication concerning this issue.

 Previous research in this field has focused on the hedging of individual reservoirs. The research aera in this paper includes external water transfer. The joint hedging rules proposed in this paper are more suitable for external water-receiving areas, which can reduce large-scale water shortage and cost of external water.

Reviewer 2 Report

In this study, the authors compared 3 different hedging policies for an efficient reservoir system operation. As case study, the Yuqiao reservoir of Tianjin (China) has been considered.

The topic and the application surely fall within the scope of MDPI Water. The paper is well structured, but its readability and clarity of exposition can be improved. Moreover, the manuscript, in its present form, contains some critical issues which need to be better discussed. Appropriate revisions to the following points should be undertaken in order to justify recommendation for publication.

1)      The authors stated that they introduced a new kind of hedging rule named JHR. I’m not sure this release curve shape has never been used in the past from other authors. There are many papers which proposed novel release curves based on the traditional HR (see for instance, https://doi.org/10.3390/w11010121 and https://doi.org/10.3390/w8060249).

I suggest to clearly state if there is some innovative contribution (e.g. in the hedging rules itself or in the PSO-based be-level optimization algorithm), if any. If not, please, insert one or more references for JHR curve shape and for the optimization algorithm.

2)      In order to enrich the bibliography, consider to include the following papers, which have recently been published on MDPI Water:

 The first and the second, describe some HR based release curves:

 Ji, Y.; Lei, X.; Cai, S.; Wang, X. Hedging Rules for Water Supply Reservoir Based on the Model of Simulation and Optimization. Water 20168, 249. https://doi.org/10.3390/w8060249

 Tayebiyan, A.; Mohammad, T.A.; Al-Ansari, N.; Malakootian, M. Comparison of Optimal Hedging Policies for Hydropower Reservoir System Operation. Water 2019, 11, 121. https://doi.org/10.3390/w11010121

 The third, consider the same problem under a different perspective. Instead of optimizing the parameters of a curve with a shape defined a priori, it adopt an artificial neural network (NNs  are known to be universal approximators) and search for its best parametrization.

 Sangiorgio, M.; Guariso, G. NN-Based Implicit Stochastic Optimization of Multi-Reservoir Systems Management. Water 2018, 10, 303. https://doi.org/10.3390/w10030303

3)      The authors said that JHR over-performs the traditional HR but looking at the results (tables 2,3,4 and tables 6,7,8) it seems that these two release curves have nearly the same performances. Please, include some results which highlights the difference between JHR and HR and clearly state it in the result/discussion/conclusion paragraphs.

4)      I suggest the authors to improve the clearness of presentation (in particular, figure 4 and table 1 need to be presented in a better way), to make the results easier to be understood (I think it is better to replace tables 2,3,4 with an histogram which helps the reader to see the differences between JHR, HR and SOP).

5)      It is not clear which data did you used for the optimization.

6)      The authors presented two different objectives (equations 7 and 8). Did you perform a single objective or a multi-objective optimization?

7)      English language and style has to be revised (see specific comments).

Specific comments:

a)      Line 17. Correct the sentence “JHR performance better” ;

b)      Line 51. “A few studies”. There are many studies in this field;

c)       Line 53. Edit “After”. The uppercase does not make sense;

d)      Line 53. “more appropriate” with respect to what?

e)      Line 63. Rephrase the sentence;

f)       Line 78. Find a better term to replace “proposed”;

g)      Line 101. Find a better term to replace “typical”. HR and SOP are used as baseline to evaluate the JHR performances;

h)      Lines 108-113. Rephrase the whole sentence;

i)        Figure 3. Pay attention to the caption;

j)        Figure 4. What does the dashed line represent? (HR or JHR)

k)      Figure 4. Try to better explain the physical meaning of the parameters (for instance, what is the physical meaning of EWA? Is it a volume?)

l)        Table 1. Reverse the order of the sections (from 1 to 5). Be consistent in the scope notation (Vt>… or …<= Vt < …).

m)    Equations 1, 2, 3. These equations should be replaced with the equation of the JHR curve (one equation for each section). In this way, one could immediately spot the quadratic form mentioned in the introduction.

n)      Many time in the text the authors wrote: “from UPPER BOUND to LOWER BOUND”. I think It is better to switch the two terms. Check all the occurrences (e.g., line 136, 139, 140, 141).

o)      The sentence at line 146-147 seems to be in contrast with the figure 5. Which is the first parameter to be optimized? SWA1 or SWA2?

p)      Why only SWAs are optimized (it seems that EWAs are not considered in the optimization process)?

q)      Line 155-157. Rephrase the sentence;

r)       Figure 5. Fix the spacing inside the boxes of the flowchart.

s)       Line 176. Replace “o” with “0”.

t)       Line 182. Find a better term to replace “life”;

u)      Line 198-199. Rephrase the sentence;

v)      Move the sentence in line 204-209 before you introduce the acronyms.

w)    Line 219. Find a better term to replace “contrast”;

x)      Line 252-253. Rephrase the sentence (“because when” does not make sense);

y)      Line 257. Figure 9 and 10 shows that the shortage occurs in day 113 and 141 approximatively. In the text the authors said that it occurs at day 250-300.

z)       Figure 10. Missing I, II, III.

aa)   Line 321. Missing figure number.

bb)  Line 337. Remove “improves HP and”

cc)    Line 337. “which considering reducing” does not make sense.

dd)  Line 342. Explain why JRC is of high reliability and practicability. Maybe these are not the best terms to use. (same issue at line 344).

ee)  Lines 348-353. Point (3) of the conclusions is not clear.

Author Response

Dear Review:

We appreciated very much to Reviewers for your positive comments, the reviewers’ comments and suggestions are very important to improve the manuscript, and the authors thank the reviewers a lot. The relative corrections were listed below.

1. The authors stated that they introduced a new kind of hedging rule named JHR. I’m not sure this release curve shape has never been used in the past from other authors. There are many papers which proposed novel release curves based on the traditional HR (see for instance, https://doi.org/10.3390/w11010121 and https://doi.org/10.3390/w8060249).

I suggest to clearly state if there is some innovative contribution (e.g. in the hedging rules itself or in the PSO-based be-level optimization algorithm), if any. If not, please, insert one or more references for JHR curve shape and for the optimization algorithm.

 The article has already cited the content of your suggestion in the text and added it to the reference. JHR is an improvement of HR in this paper. No other literature has discussed the curve shape of JHR. Some researchers have used optimization algorithms to study HR, and this article has added relevant references.

2. In order to enrich the bibliography, consider to include the following papers, which have recently been published on MDPI Water:

The first and the second, describe some HR based release curves:Ji, Y.; Lei, X.; Cai, S.; Wang, X. Hedging Rules for Water Supply Reservoir Based on the Model of Simulation and Optimization. Water 2016, 8, 249. https://doi.org/10.3390/w8060249

Tayebiyan, A.; Mohammad, T.A.; Al-Ansari, N.; Malakootian, M. Comparison of Optimal Hedging Policies for Hydropower Reservoir System Operation. Water 2019, 11, 121. https://doi.org/10.3390/w11010121

The third, consider the same problem under a different perspective. Instead of optimizing the parameters of a curve with a shape defined a priori, it adopt an artificial neural network (NNs are known to be universal approximators) and search for its best parametrization. Sangiorgio, M.; Guariso, G. NN-Based Implicit Stochastic Optimization of Multi-Reservoir Systems Management. Water 2018, 10, 303. https://doi.org/10.3390/w10030303

 The article has already cited the content of your suggestion in the text and added it to the reference.

3. The authors said that JHR over-performs the traditional HR but looking at the results (tables 2,3,4 and tables 6,7,8) it seems that these two release curves have nearly the same performances. Please, include some results which highlights the difference between JHR and HR and clearly state it in the result/discussion/conclusion paragraphs.

 Through the comparison of Table 2 and Table 3, in the case of (d, d), (m, d), (w, d) under the condition of JHR has less transfer water transfer, which reduces the water transfer cost. The water supply is small when the external water transfer is extremely dr. Therefore, the paper mainly compares the number of days in different water shortage rate under the four conditions. Tables 5 to 8 are the results of reducing the number of days of different water shortage rates under the condition of JHR, HR and SOP. The results show that JHR and HR can reduce the occurrence of severe water shortage in the absence of water (Table 5~ Table 8 does not show that JHR is better than HR which was shown in Tables 2, 3, and 4. Smaller water diversion under JHR can reduce water diversion costs).

4. I suggest the authors to improve the clearness of presentation (in particular, figure 4 and table 1 need to be presented in a better way), to make the results easier to be understood (I think it is better to replace tables 2,3,4 with a histogram which helps the reader to see the differences between JHR, HR and SOP).

 In figure 4, JRH was mistakenly written as RH and has been corrected. Table 1 has been modified according to the suggestion in the specific comments. As for table 2-4, I added a histogram (Figure 9) to better compare the different results of JRH, RH, and sop and make the results easier to be understood. And the two pictures show the advantages of RH relative to SOP and the advantage of JRH relative to RH respectively.

5. It is not clear which data did you used for the optimization.

 This paper collected daily runoff data of Yuqiao Reservoir and the Danjiangkou reservoir, as for SWNDP, according to Proximity Principle in runoff data, 1981 and 1987 are respectively dry year and extremely dry year; 1998, 1984, 1982 and 2000 are respectively Yuqiao Reservoir’s wet year, normal year, dry year and extremely dry year. The water demand is the average of the water consumption in Tianjin in the past ten years. This paper used the runoff data of above years and water demand for optimization.

6. The authors presented two different objectives (equations 7 and 8). Did you perform a single objective or a multi-objective optimization?

 The two-layer model adopted in this paper, the upper and lower models compete with each other to find the optimal solution. The optimal solution is obtained in the upper model, which is used as the input of the lower layer model. Then the optimal solution is substituted into the upper model in the lower model, and the cycle is calculated back and forth until the condition of the optimal solution is satisfied.

7. English language and style has to be revised (see specific comments).

a) Line 17. Correct the sentence “JHR performance better”;

 Accepted.

b) Line 51. “A few studies”. There are many studies in this field;

 Have changed it to “several studies”.

c) Line 53. Edit “After”. The uppercase does not make sense;

 Accepted and deleted the sentence after “after”.

d) Line 53. “more appropriate” with respect to what?

 It means that the developed hedge rules are more practical. But the sentence does not make sense and has been deleted.

e) Line 63. Rephrase the sentence;

 Accepted and have rephrase the sentence.

f) Line 78. Find a better term to replace “proposed”;

 “proposed” has been rewritten into “put forward”.

g) Line 101. Find a better term to replace “typical”. HR and SOP are used as baseline to evaluate the JHR performances;

 “Typical” has been rewritten as “representative” and Insert the sentence “HR and SOP are used as baseline to evaluate the JHR performances”.

h) Lines 108-113. Rephrase the whole sentence;

 Accepted and rephrase the whole sentence.

i) Figure 3. Pay attention to the caption;

 Accepted

j) Figure 4. What does the dashed line represent? (HR or JHR)

 I am sorry. The dashed line represented JHR. The mistake has been corrected.

k) Figure 4. Try to better explain the physical meaning of the parameters (for instance, what is the physical meaning of EWA? Is it a volume?)

 Have added a sentence to explain the physical meaning of the parameters. SWA and EWA are the reservoir storage capacity above the dead water level of the reservoir, which is also the water supply capacity of the reservoir in the paper.

l) Table 1. Reverse the order of the sections (from 1 to 5). Be consistent in the scope notation (Vt>… or …<= Vt < …).

Thank you for your reminder. Accepted.

m) Equations 1, 2, 3. These equations should be replaced with the equation of the JHR curve (one equation for each section). In this way, one could immediately spot the quadratic form mentioned in the introduction.

Thank you for your suggestion. Equations 1, 2, 3. are used to calculate the paraments p and q. The two paraments are set for section 2, 4, 5, so this paper only listed 3 equations. In section 1, reservoir supplies water according to water supply capacity. In section 3, Mt and Lt are the part which needed to be satisfied by the reservoir and the transferred water in t period and has nothing to do with the two parameters.

n) Many times in the text the authors wrote: “from UPPER BOUND to LOWER BOUND”. I think It is better to switch the two terms. Check all the occurrences (e.g., line 136, 139, 140, 141).

 Accepted.

o) The sentence at line 146-147 seems to be in contrast with the figure 5. Which is the first parameter to be optimized? SWA1 or SWA2?

 The optimization of SWA1 and SWA2 is carried out at the same time. The optimization process of these two parameters affect each other and belongs to the Starkberg game.

p) Why only SWAs are optimized (it seems that EWAs are not considered in the optimization process)?

 SWA1 and SWA2 are optimized solutions. EWA2 is the water storage capacity corresponding to the normal water storage level in the dry season and corresponding to the flood control limit water level in the wet season. It is the parameter of the reservoir and does not need to be optimized; and EWA1=qs*SWA1 in this paper. This setting is to reduce the difficulty of optimizing the solution.

q) Line 155-157. Rephrase the sentence;

Accepted and have rephrased the sentence.

r) Figure 5. Fix the spacing inside the boxes of the flowchart.

 Accepted

s) Line 176. Replace “o” with “0”.

Accepted.

t) Line 182. Find a better term to replace “life”;

Have changed it

u) Line 198-199. Rephrase the sentence;

Accepted.

v) Move the sentence in line 204-209 before you introduce the acronyms.

Accepted.

w) Line 219. Find a better term to replace “contrast”;

Accepted.

x) Line 252-253. Rephrase the sentence (“because when” does not make sense);

Accepted and deleted it.

y) Line 257. Figure 9 and 10 shows that the shortage occurs in day 113 and 141 approximatively. In the text the authors said that it occurs at day 250-300.

 Thank you for your attention. Accepted and have rewritten it. It means that large scale shortage occurs at day 250-300.

z)  Figure 10. Missing I, II, III.

Accepted.

aa) Line 321. Missing figure number.

Thank you for your reminder. Have added it.

bb) Line 337. Remove “improves HP and”

Accepted

cc) Line 337. “which considering reducing” does not make sense.

Accepted and deleted it.

dd) Line 342. Explain why JRC is of high reliability and practicability. Maybe these are not the best terms to use. (same issue at line 344).

The expression is not accurate enough. The meaning of this sentence is that JHR can reduce the risk of serious water shortage events and the cost of external water diversion.

ee) Lines 348-353. Point (3) of the conclusions is not clear.

 Have rewritten it.

Round  2

Reviewer 1 Report

Authors did not present the version of the manuscript beforeresubmission, the whole references are marked as a new one. Figures are illegible, it was not improved. Novelty of the manuscript should be underlined.

Author Response

Dear Review:

We appreciated very much to Reviewers for your positive comments, the reviewers’ comments and suggestions are very important to improve the manuscript, and the authors thank the reviewers a lot. The relative corrections were listed below.

 Authors did not present the version of the manuscript before resubmission; the whole references are marked as a new one. Figures are illegible, it was not improved. Novelty of the manuscript should be underlined.

 We are sorry to forget to present the version of the manuscript before resubmission. The version of the manuscript before resubmission has been submitted this time. Because the reference is generated using a macro, a new macro is generated for each modification. We have made the figures more legible and underline the novelty of the manuscript in line 95-103.

Reviewer 2 Report

2. In order to enrich the bibliography, consider to include the following papers, which have recently been published on MDPI Water:

The first and the second, describe some HR based release curves:Ji, Y.; Lei, X.; Cai, S.; Wang, X. Hedging Rules for Water Supply Reservoir Based on the Model of Simulation and Optimization. Water 2016, 8, 249. https://doi.org/10.3390/w8060249

Tayebiyan, A.; Mohammad, T.A.; Al-Ansari, N.; Malakootian, M. Comparison of Optimal Hedging Policies for Hydropower Reservoir System Operation. Water 2019, 11, 121. https://doi.org/10.3390/w11010121

The third, consider the same problem under a different perspective. Instead of optimizing the parameters of a curve with a shape defined a priori, it adopt an artificial neural network (NNs are known to be universal approximators) and search for its best parametrization. Sangiorgio, M.; Guariso, G. NN-Based Implicit Stochastic Optimization of Multi-Reservoir Systems Management. Water 2018, 10, 303. https://doi.org/10.3390/w10030303

 ----

The article has already cited the content of your suggestion in the text and added it to the reference.

----

I would stress the concept expressed in the third reference listed in my comment:
The third, consider the same problem under a different perspective. Instead of optimizing the parameters of a curve with a shape defined a priori, it adopt an artificial neural network (NNs are known to be universal approximators) and search for its best parametrization. Sangiorgio, M.; Guariso, G. NN-Based Implicit Stochastic Optimization of Multi-Reservoir Systems Management. Water 2018, 10, 303. https://doi.org/10.3390/w10030303

--------------------------------------------------------------------------------------------------------------------------

6. The authors presented two different objectives (equations 7 and 8). Did you perform a single objective or a multi-objective optimization?

----

The two-layer model adopted in this paper, the upper and lower models compete with each other to find the optimal solution. The optimal solution is obtained in the upper model, which is used as the input of the lower layer model. Then the optimal solution is substituted into the upper model in the lower model, and the cycle is calculated back and forth until the condition of the optimal solution is satisfied.

----

It is better to underline that the presented optimization method can deal with multi objective optimization problems.

--------------------------------------------------------------------------------------------------------------------------

a) Line 17. Correct the sentence “JHR performance better”;

----

 Accepted.

----

Performance is not a verb. You have to use the verb “to perform”.

--------------------------------------------------------------------------------------------------------------------------

cc) Line 337. “which considering reducing” does not make sense.

----

Accepted and deleted it.

----

I think you have to correct English. Use “which+present simple” insread of “which + -ing”. There are still some error in English style and language.

--------------------------------------------------------------------------------------------------------------------------

The reference numbers has to be fixed (pay attention at the number 13rd of the reference list)

--------------------------------------------------------------------------------------------------------------------------

The legend of figure 7 has been cut. Fix it and try to strongly reduce the dimension of this figure since it takes a while to visualize it on the screen.

--------------------------------------------------------------------------------------------------------------------------

Use the same green color for all the figures 10-11-12-13

Author Response

Dear Review:

We appreciated very much to Reviewers for your positive comments, the reviewers’ comments and suggestions are very important to improve the manuscript, and the authors thank the reviewers a lot. The relative corrections were listed below.

I would stress the concept expressed in the third reference listed in my comment:

The third, consider the same problem under a different perspective. Instead of optimizing the parameters of a curve with a shape defined a priori, it adopts an artificial neural network (NNs are known to be universal approximators) and search for its best parametrization. Sangiorgio, M.; Guariso, G. NN-Based Implicit Stochastic Optimization of Multi-Reservoir Systems Management. Water 2018, 10, 303. https://doi.org/10.3390/w10030303

We have revised the reference to this article in the paper.

The authors presented two different objectives (equations 7 and 8). Did you perform a single objective or a multi-objective optimization? It is better to underline that the presented optimization method can deal with multi objective optimization problems.

We have underline that the presented optimization method can deal with multi objective optimization problems in line 162-165.

Line 17. Correct the sentence “JHR performance better”; Performance is not a verb. You have to use the verb “to perform”.

We have change “JHR performance better” to “JHR performs better”.

Line 337. “which considering reducing” does not make sense. I think you have to correct English. Use “which+present simple” insread of “which + -ing”. There is still some error in English style and language.

We have corrected the error in the sentence.

The reference numbers have to be fixed (pay attention at the number 13rd of the reference list)

We have corrected the reference.

The legend of figure 7 has been cut. Fix it and try to strongly reduce the dimension of this figure since it takes a while to visualize it on the screen.

The legend is in the lower left corner of the figure 7. We have reduced the dimension of this figure.

Use the same green color for all the figures 10-11-12-13

Accepted and used the same green color for all the figures 10-11-12-13.

Round  3

Reviewer 1 Report

Figures should be of better quality.

English language and style should be checked and revised.